

# Comprehensive circular RNA expression profile in radiation-treated HeLa cells and analysis of radioresistance-related circRNAs

Duo Yu[1], Yunfeng Li[1], Zhihui Ming[2], Hongyong Wang[1], Zhuo Dong[3], Ling Qiu[1] and Tiejun Wang[1]

[1] Radiotherapy Department, 2nd Hospital Affiliated to Jilin University, Changchun, China
[2] Stomatology Department, 1st Hospital Affiliated to Jilin University, Changchun, China
[3] College of Public Medicine, Key Laboratory of Radiobiology, Ministry of Health, Jilin University, Changchun, China

Corresponding author
Tiejun Wang,
wangtiejunyd@sina.com

## ABSTRACT

**Background:** Cervical cancer is one of the most common cancers in women worldwide. Malignant tumors develop resistance mechanisms and are less sensitive to or do not respond to irradiation. With the development of high-throughput sequencing technologies, circular RNA (circRNA) has been identified in an increasing number of diseases, especially cancers. It has been reported that circRNA can compete with microRNAs (miRNAs) to change the stability or translation of target RNAs, thus regulating gene expression at the transcriptional level. However, the role of circRNAs in cervical cancer and the radioresistance mechanisms of HeLa cells are unknown. The objective of this study is to investigate the role of circRNAs in radioresistance in HeLa cells.

**Methods:** High-throughput sequencing and bioinformatics analysis of irradiated and sham-irradiated HeLa cells. The reliability of high-throughput RNA sequencing was validated using quantitative real-time polymerase chain reaction. The most significant circRNA functions and pathways were selected by Gene Ontology (GO) and Kyoto Encyclopedia of Genes and Genomes (KEGG) analyses. A circRNA–miRNA–target gene interaction network was used to find circRNAs associated with radioresistance. Moreover, a protein–protein interaction network was constructed to identify radioresistance-related hub proteins.

**Results:** High-throughput sequencing allowed the identification of 16,893 circRNAs involved in the response of HeLa cells to radiation. Compared with the control group, there were 153 differentially expressed circRNAs, of which 76 were up-regulated and 77 were down-regulated. GO covered three domains: biological process (BP), cellular component (CC) and molecular function (MF). The terms assigned to the BP domain were peptidyl-tyrosine dephosphorylation and regulation of cell migration. The identified CC terms were cell–cell adherens junction, nucleoplasm and cytosol, and the identified MF terms were protein binding and protein tyrosine phosphatase activity. The top five KEGG pathways were MAPK signaling pathway, endocytosis, axon guidance, neurotrophin signaling pathway, and SNARE interactions in vesicular transport. The protein–protein interaction analysis indicated that 19 proteins might be hub proteins.

**Conclusions:** CircRNAs may play a major role in the response to radiation. These findings may improve our understanding of the role of circRNAs in radioresistance in HeLa cells and allow the development of novel therapeutic approaches.

## INTRODUCTION

Cervical cancer is one of the most common cancers in women worldwide and is responsible for the high mortality of female cancers (*Du et al., 2012*). In recent years, with improvements in radiotherapy, radiochemotherapy after operation is the standard treatment for cervical cancer. However, recurrence and metastasis after radiotherapy remain a major problem in the treatment of locally-advanced cervical cancer (*De Freitas, Gomes & Coimbra, 2015*; *Zhao et al., 2013*). Therefore, it is crucial to understand the mechanisms underlying the development of radioresistance to improve the therapeutic effect of radiotherapy in the treatment of cervical cancer.

Circular RNAs (circRNAs) are non-coding RNA molecules that compose a RNA class that forms covalent closed-loop structures without 5′-3′polarity (*Qu et al., 2015*). The development of novel biochemical and computational approaches has focused RNA research on circRNAs. CircRNAs are involved in many human diseases, including cancer (*Burd et al., 2010*; *Bachmayr-Heyda et al., 2015*). Some studies have demonstrated that circRNAs play a major role in the biological function of a network of competing endogenous RNAs. Importantly, circRNAs may compete with micro RNAs (miRNAs) to change the stability and translation of target RNAs, thus regulating gene expression at the transcriptional level (*Bachmayr-Heyda et al., 2015*; *Zhong et al., 2018*; *Ivanov et al., 2015*; *Qu et al., 2015*). Despite this significant progress, the expression profile, mechanism of action, and biological activity of circRNA have not been completely elucidated.

In our study, high-throughput RNA sequencing was used to investigate the differential expression profiles of circRNAs between irradiated HeLa cells and sham-irradiated HeLa cells. The comprehensive analysis of radioresistance-related circRNAs may lay a foundation for future studies on the diagnosis and treatment of cervical cancer.

## MATERIALS AND METHODS

### HeLa cells and irradiation

HeLa cells were purchased from Procell Life Science and Technology Co. Ltd. (Wuhan, China) with short tandem repeat qualification report (Supplementary Material). The cells were cultured in MEM medium (Gibco; Life Technologies Inc., Grand Island, NY, USA) supplemented with 10% fetal bovine serum (Sigma), 1% antibiotic-antimitotic solution (Gibco; Life Technologies, Grand Island, NY, USA). An X-ray generator (Model X-RAD320i X; Precision X-ray, Inc., North Branford, CT, USA) was used to deliver radiation at a dose rate of 1.020 Gy/min (180 kV; 20 mA) for a total dose of 10 Gy in the treatment group (irradiated). The control group (sham-irradiated) did not receive radiation as blank comparison. Each group had three samples.

## High-throughput sequencing of circRNAs and differential expression analysis

Total RNA was extracted from each group by using TRIzol reagent (Invitrogen, Carlsbad, CA, USA) following the manufacturer's protocol. RNA quantity and quality were determined in a Nano Drop ND-1000 spectrophotometer (Nano Drop Technologies, Inc., Wilmington, DE, USA). RNA integrity was evaluated by standard denaturing agarose gel electrophoresis using the Agilent 2100 Bio analyzer. The total RNA was measured in a spectrophotometer at the wavelengths of 260 nm and 280 nm. The samples with an OD 260/280 ratio of approximately 2.0 were used. CircRNAs were quantitatively analyzed by Novogene Bioinformatics Technology Co. Ltd. (Beijing, China). After removal of ribosomal RNA and building a library, high-throughput RNA sequencing was performed. Find_circ was used to identify circRNAs (*Memczak et al., 2013*). Differentially expressed circRNAs were detected by the negative binomial distribution test using the DESeq2 package. Two criteria were chosen: (i) $|\log_2(\text{foldchange})|>1$ and (ii) *p*-value ($p < 0.05$).

## Construction of the circRNA–miRNA–target gene interaction network

Mounting evidence has indicated that circRNAs may regulate the activity of miRNAs by acting as competing endogenous RNAs or miRNA sponges. The role of circRNAs in HeLa cells subjected to radiation was determined by building a circRNA–miRNA–target gene co-expression network. We used the top four down-regulated and top four up-regulated circRNAs to construct this network using Cytoscape software version 3.2.1. Putative interactions between miRNAs and circRNAs were evaluated using miRanda (3.3a). The top five (highest targeting relationship score) miRNAs were selected, and the target genes of these miRNAs were predicted using miRDB (http://www.mirdb.org). Target scores > 98 were selected to construct the circRNA–miRNA–mRNA interaction network.

## Gene ontology and kyoto encyclopedia of genes and genomes pathway analysis

The gene ontology (GO) analysis provides a controlled terminology to describe gene and gene product attributes for many organisms. In this study, DAVID (https://david.ncifcrf.gov/) was used to analyze the potential function of target genes. *p*-values < 0.05 were considered statistically significant. GO covers three domains: biological process (BP), cellular component (CC), and molecular function (MF). The five most enriched GO terms were ranked by *p*-value. Kyoto encyclopedia of genes and genomes (KEGG) pathway analysis was conducted to determine the involvement of target genes in different biological pathways. DAVID (https://david.ncifcrf.gov/) was also used in this part of the analysis.

## Protein–protein interaction network analysis

The online tool STRING (https://string-db.org) was used to analyze the target genes. The selection criterion was the highest confidence (interaction score > 0.9). Cytoscape software version 3.2.1 was used to draw the protein–protein interaction (PPI) network.

The hub proteins were selected by their association with other proteins. The target genes with more associations have important roles in the PPI interaction network.

## Analysis of the expression level of circRNAs using qRT-PCR

The reliability of high-throughput RNA sequencing was validated. The top four up-regulated circRNAs and top four down-regulated circRNAs were selected. Quantitative real-time polymerase chain reaction (qRT-PCR) was performed using Q SYBR green Supermix (Bio Rad, Hercules, CA, USA), and PCR-specific amplification was conducted in the 7900 HT Sequence Detection System (ABI PRISM; Waltham, MA, USA). The expression was determined by using the threshold cycle (Ct), and relative expression levels were calculated via the $2^{-\Delta\Delta Ct}$ method. *GAPDH* served as the standard internal control and all reactions were performed in triplicate.

# RESULTS

## Expression pattern of circRNAs during the priming phase of radiation-treated HeLa cells

High-throughput sequencing was used to determine the expression profile of circRNAs. Two groups of HeLa cells were used, and each group contained three samples. The correlation between gene expression levels in different samples indicated the biological repetition (Fig. 1A). The analysis of sequencing data allowed identifying 16,893 circRNAs in HeLa cells. The length of HeLa circRNA candidates ranged from <150 to 99,934 nucleotides (nt). Approximately 67% of circRNAs had the predicted spliced length of <10,000 nt, whereas 46.4% and 32.0% of the circRNAs had a length of <5,000 and 10,000–50,000 nt, respectively (Fig. 1B). Among the 16,893 circRNAs, 11,456 were detected in the control group, 11,018 were detected in the treatment group, and 5,581 were detected in both groups. The hierarchical clustering showed the gene expression patterns of the samples (Figs. 1C and 1D).

## Identification of differentially-expressed circRNAs in HeLa cells

The |log$_2$ (foldchange)|>1 and *p*-value ($p < 0.05$) were used to evaluate significant differences in the expressions of circRNAs between the two groups. A total of 153 circRNAs were differentially expressed in the Treatment Group compared with the Control Group. Of these, the volcano plot indicated that 76 circRNAs were up-regulated and 77 were down-regulated (Fig. 1E). The novel genomic feature of circRNAs in both groups is shown in Fig. 1F.

## Validation of circRNA expression

To verify the high-throughput sequencing results, the top four up-regulated and top four down-regulated circRNAs were selected and their expression levels were validated by qRT-PCR analysis. The qRT-PCR results are shown in Fig. 2. hsa_circ_0009035, hsa_circ_0000392, hg38_circ_0004913, hsa_circ_0004015 were significantly up-regulated in the Treatment Group compared with the Control Group, whereas hg38_circ_0013682, hg38_circ_0015954, hsa_circ_0013738, and hsa_circ_0013225 were down-regulated in

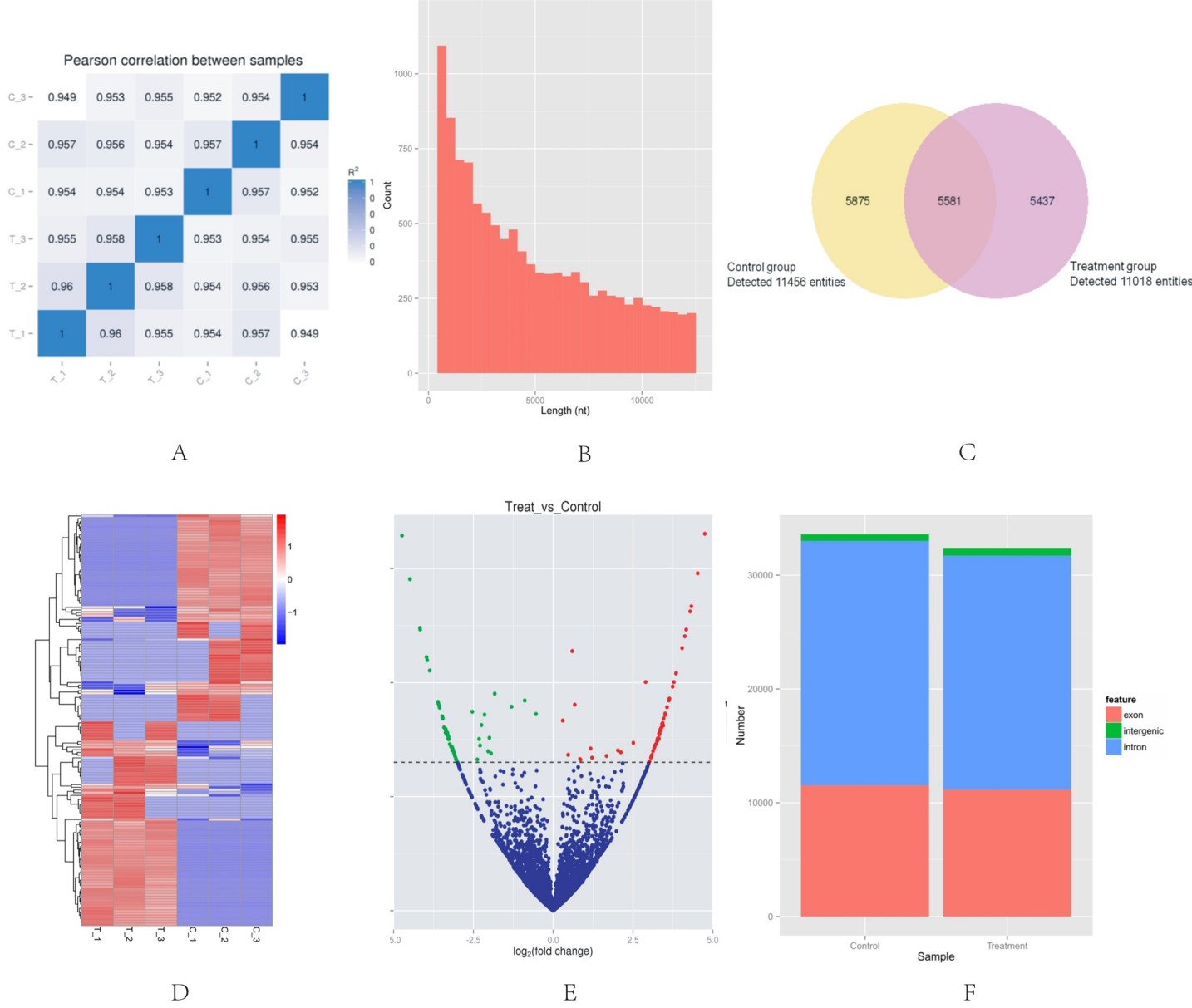

**Figure 1 Basic situation of high-throughput sequencing.** (A) Inter–sample expression correlation test. Two groups of HeLa cells were used and each group contained three samples. The cell groups were treated by different methods. The correlation between gene expression levels in different samples indicated the biological repetition. The greater the absolute value of *r* indicates that the stronger the correlation. (B) CircRNA length distribution. Approximately 67% of circRNAs had the predicted spliced length of <10,000 nt, whereas 46.4% and 32.0% of the circRNAs had a length of <5,000 and 10,000–50,000 nt, respectively. (C) Venn diagram of the differential expression of circRNAs. The yellow part of the circle is the Control group and the purple part of the circle is the Treatment group. The total number of the circRNAs was 16,893. Approximately 11,456 circRNAs were detected in the control group, 11,018 in the treatment group, and 5,581 in both groups. (D) Hclusterheatmap. Hierarchical clustering showing the differential expression profile of circRNAs between the two study groups and the homogeneity within each group. (E) Volcano plot of differentially expressed circRNAs. The green and red dots in the plot represent the differentially expressed circRNAs with statistical significance. The red dots correspond to upregulated circRNAs and the green dots correspond to downregulated circRNAs. (F) circRNA genomic feature data. It shows the novel genomic feature of circRNAs in both groups. The red part is exon, green part is intergenic, and blue part is intron.

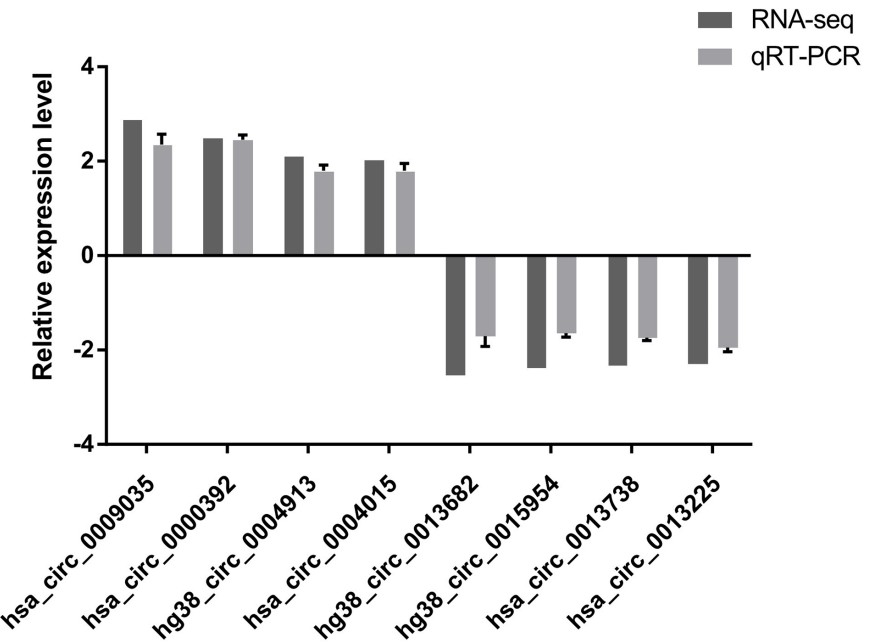

**Figure 2  qRT-PCR results.** The expression patterns of the top four up-regulated and top four down-regulated circRNAs as monitored by qRT-PCR. The expression patterns of these eight circRNAs determined by qRT-PCR agreed with the results of the high-throughput sequencing analysis and demonstrated the reliability of the RNA high-throughput sequencing technology.

the Treatment Group. The expression patterns of these eight circRNAs determined by qRT-PCR agreed with the results of the high-throughput sequencing analysis. The sequence of the primers used in qRT-PCR is shown in Table 1.

## Construction of the circRNA–miRNA–target gene interaction network

A circRNA–miRNA–target gene co-expression network was constructed to evaluate the potential functions of circRNAs. Cytoscape was used to draw the circRNA–miRNA–target gene interaction network (Fig. 3). It is of interest that circRNAs may play a central role in this interaction network, and a single circRNA may correlate with many miRNAs and regulate many target genes. This network may help elucidate the potential function of circRNAs and their mechanism of action.

## GO and KEGG analysis of target genes

The role of circRNAs in response to radiation was investigated by performing a GO functional analysis. The target genes from the circRNA–miRNA–target gene interaction network were used in this analysis. GO has three different domains: BP, CC, and MF. The top five (ranked by $p$-value) BP terms, CC terms, and MF terms are shown in Fig. 4 and Table 2. The identified BP terms were peptidyl-tyrosine dephosphorylation, regulation of cell migration, positive regulation of transcription from RNA polymerase II promoter, angiogenesis, and morphogenesis of an epithelial sheet. The identified CC terms were cell–cell adherens junction, nucleoplasm, cytosol, lamellipodium, and nucleus. The identified MF terms were protein binding, tyrosine phosphatase activity, cadherin binding

**Table 1  Primers used in qRT-PCR.**

| Gene | Forward primer (5′ to 3′) | Reverse primer (5′ to 3′) | Variation tendency |
|---|---|---|---|
| Gapdh | TGACTTCAACAGCGACACCCA | CACCCTGTTGCTGTAGCCAAA | |
| hg38_circ_0013682 | GGCAGACAGAAGGAAACAGC | GCTTTTGCTCTTGGGTTCTG | down-regulated |
| hg38_circ_0015954 | CAAGGACTGCCTGATTGACAAG | GGAGGTGAGGGAGGAGTTCA | down-regulated |
| hsa_circ_0013738 | TCCTTCCTGCCTTTAACACAC | TGGTAGCACCCATTTGTGAA | down-regulated |
| hsa_circ_0013225 | CCGGACACTTGTTTTCCAGT | TTCTGTTTGTGAGCAATCATCC | down-regulated |
| hg38_circ_0004913 | CTGCCATAGGACAGGCTGA | GGCACAAAGACAGCCTAATGA | up-regulated |
| hsa_circ_0009035 | TTAGGTGGTTGAGCGCCTGC | GGGCAGTTCACCAACAGCTT | up-regulated |
| hsa_circ_0000392 | ACAGAAGGGCAAGAGAGGTGG | TTCCTTGGTCCTCGAGGCAC | up-regulated |
| hsa_circ_0004015 | AGGGGAAGGATCTTATGCTACAGT | CACTGAGTCCATTCCCTGGCA | up-regulated |

involved in cell–cell adhesion, RNA polymerase II core promoter proximal region sequence-specific DNA binding, transcriptional activator activity, and RNA polymerase II core promoter proximal region sequence-specific binding. The top five (ranked by gene count) KEGG pathways are shown in Fig. 5 and Table 3. The identified pathways were MAPK signaling, endocytosis, axon guidance, neurotrophin signaling, and SNARE interactions in vesicular transport. It is of note that five genes—*RPS6KA5, RPS6KA6, CRKL, RAP1A, FASLG, MAPK8*—were found in both the MAPK and neurotrophin signaling pathway.

## PPI network

STRING was used to predict protein interactions between the target genes. Interaction scores higher than 0.9 (highest confidence) were selected for constructing PPI networks (Fig. 6). Nineteen proteins—IGF2R, DNAJC6, FZD4, CBL, CLTC, ARPC3, VAMP2, ITGB3, VAMP4, ITGB1, MAPK8, KIF2A, CLASP1, SMC3, STAG2, STAG1, H2AFZ, RBBP4, and UBE2D3—were strongly correlated with other proteins (connected with >10 proteins). These hub proteins might play a crucial role in the radioresistance of cervical cancer.

## DISCUSSION

The development of high-throughput sequencing technology has increased our understanding of the role of circRNAs. A recent study found that circRNAs are important members of the non-coding RNA family in different species. The role of circRNA in human disease, especially in cancer, is vital. The functions and biological characteristics of circRNAs may help elucidate the mechanisms underlying human disease. Radioresistance is considered a severe complication of cervical cancer radiotherapy, and the radioresistance mechanism remains poorly understood. In this study, circRNA sequencing was used to analyze the circRNA expression profile between irradiated HeLa cells and sham-irradiated HeLa cells and explore the possible involvement of differentially-expressed circRNAs in radioresistance mechanisms.

In the present study, a 10 Gy X-ray was applied to HeLa cells, which were collected 48 h later. This interval was chosen because most highly-expressed radiation resistance-related genes are strongly inhibited by radiation after 48 h (*Shen et al., 2017b*; *Ghosh & Krishna, 2012*; *Balkwill, 2004*; *Zhang et al., 2017*). High-throughput RNA sequencing allowed the
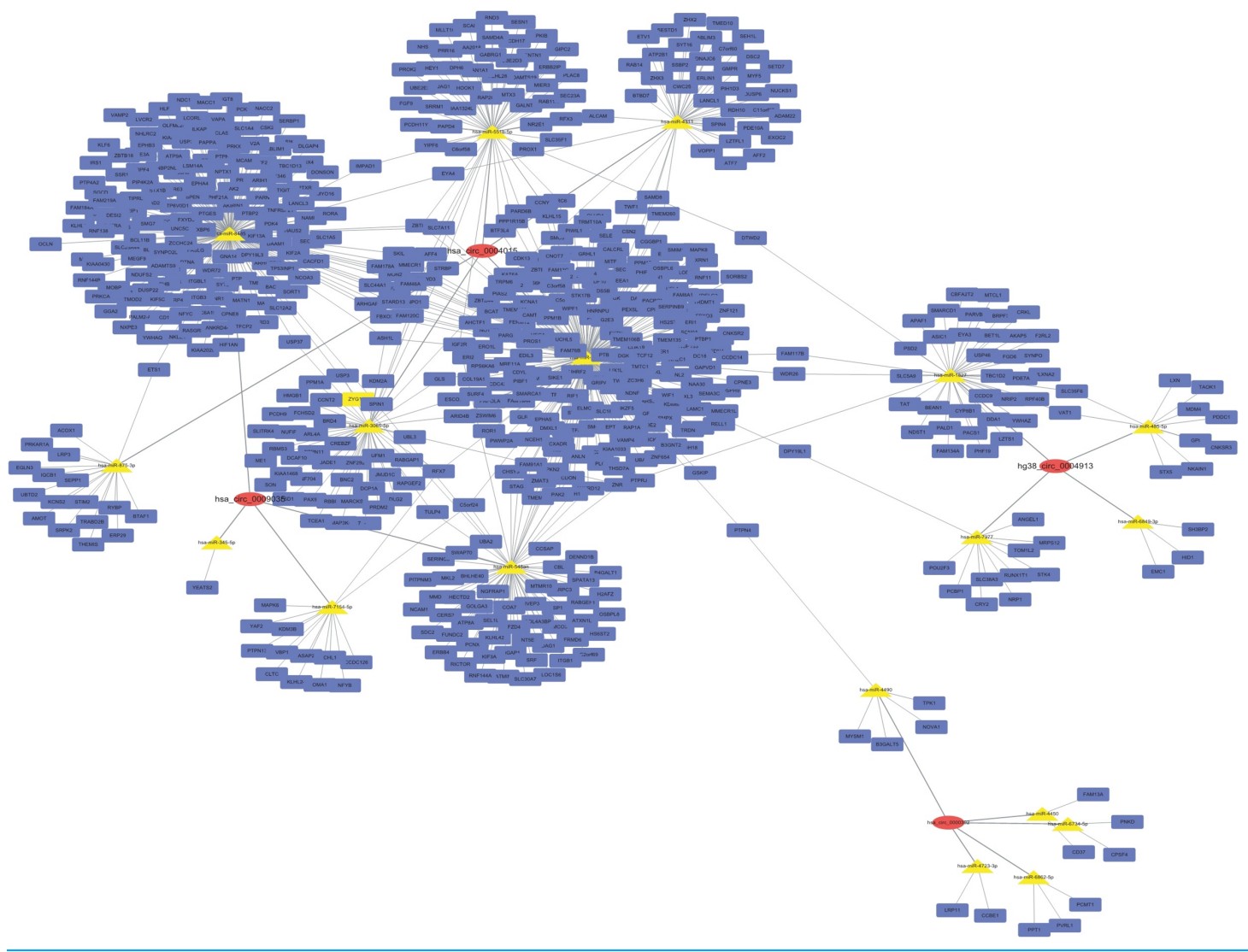

**Figure 3** **CircRNA–miRNA–target gene interaction network.** The top four down- and up-regulated circRNAs were used to construct this network using Cytoscape. Putative interactions between miRNAs and circRNAs were evaluated using miRanda. The top five (highest targeting relationship score) miRNAs were selected, then the same miRNAs' target genes were predicted using miRDB. Target scores > 98 were selected. In this figure, rectangles represent mRNAs while ovals represent circRNAs and triangles represent miRNAs.

identification of 16,893 circRNAs in both Treatment and Control Group. A total of 153 circRNAs was differentially expressed ($p < 0.05$, |$\log_2$(foldchange)|>1). Four up-regulated and four down-regulated circRNAs validated by qRT-PCR were significantly dysregulated, and this result is consistent with the high-throughput sequencing, indicating the high reliability of high-throughput sequencing data.

A circRNA–miRNA–target gene interaction network was constructed to further investigate the regulatory role of circRNAs in radioresistance. This network showed that circRNAs might play a central regulatory role. A single circRNA may be associated with many miRNAs and then may regulate more target genes. For instance, in our study, hsa_circ_0004015 may bind to hsa-miR-3163, hsa-miR-3065-5p, hsa-miR-551b-5p,

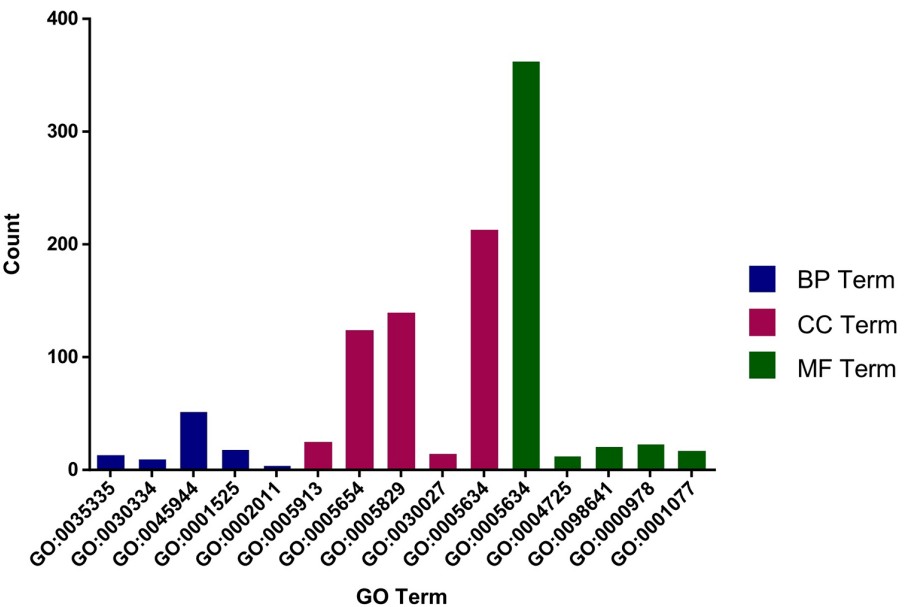

**Figure 4 Top five GO terms from BP, CC and MF.** The top five GO terms in each group were ranked by *p*-value. They are peptidyl-tyrosine dephosphorylation; regulation of cell migration; positive regulation of transcription from RNA polymerase II promoter; angiogenesis; morphogenesis of an epithelial sheet (from BP). Cell–cell adherens junction; nucleoplasm; cytosol; lamellipodium; nucleus (from CC). Protein binding; protein tyrosine phosphatase activity; cadherin binding involved in cell–cell adhesion; RNA polymerase II core promoter proximal region sequence-specific DNA binding; transcriptional activator activity, RNA polymerase II core promoter proximal region sequence-specific binding (from MF).

**Table 2 The top five (ranked by *p*-value) BP terms, CC terms, and MF terms.**

| GO | *p*-value | Description | GO terms |
|---|---|---|---|
| GO:0035335 | 1.20E-04 | Peptidyl-tyrosine dephosphorylation | Biological_process |
| GO:0030334 | 8.18E-04 | Regulation of cell migration | Biological_process |
| GO:0045944 | 0.0013274 | Positive regulation of transcription from RNA polymerase II promoter | Biological_process |
| GO:0001525 | 0.0014863 | Angiogenesis | Biological_process |
| GO:0002011 | 0.0018697 | Morphogenesis of an epithelial sheet | Biological_process |
| GO:0005913 | 1.67E-04 | Cell-cell adherens junction | Cellular_component |
| GO:0005654 | 2.11E-04 | Nucleoplasm | Cellular_component |
| GO:0005829 | 7.43E-04 | Cytosol | Cellular_component |
| GO:0030027 | 7.79E-04 | Lamellipodium | Cellular_component |
| GO:0005634 | 9.11E-04 | Nucleus | Cellular_component |
| GO:0005515 | 6.07E-10 | Protein binding | Molecular_function |
| GO:0004725 | 4.76E-04 | Protein tyrosine phosphatase activity | Molecular_function |
| GO:0098641 | 0.001694 | Cadherin binding involved in cell-cell adhesion | Molecular_function |
| GO:0000978 | 0.003848 | RNA polymerase II core promoter proximal region sequence-specific DNA binding | Molecular_function |
| GO:0001077 | 0.005654 | Transcriptional activator activity, RNA polymerase II core promoter proximal region sequence-specific binding | Molecular_function |

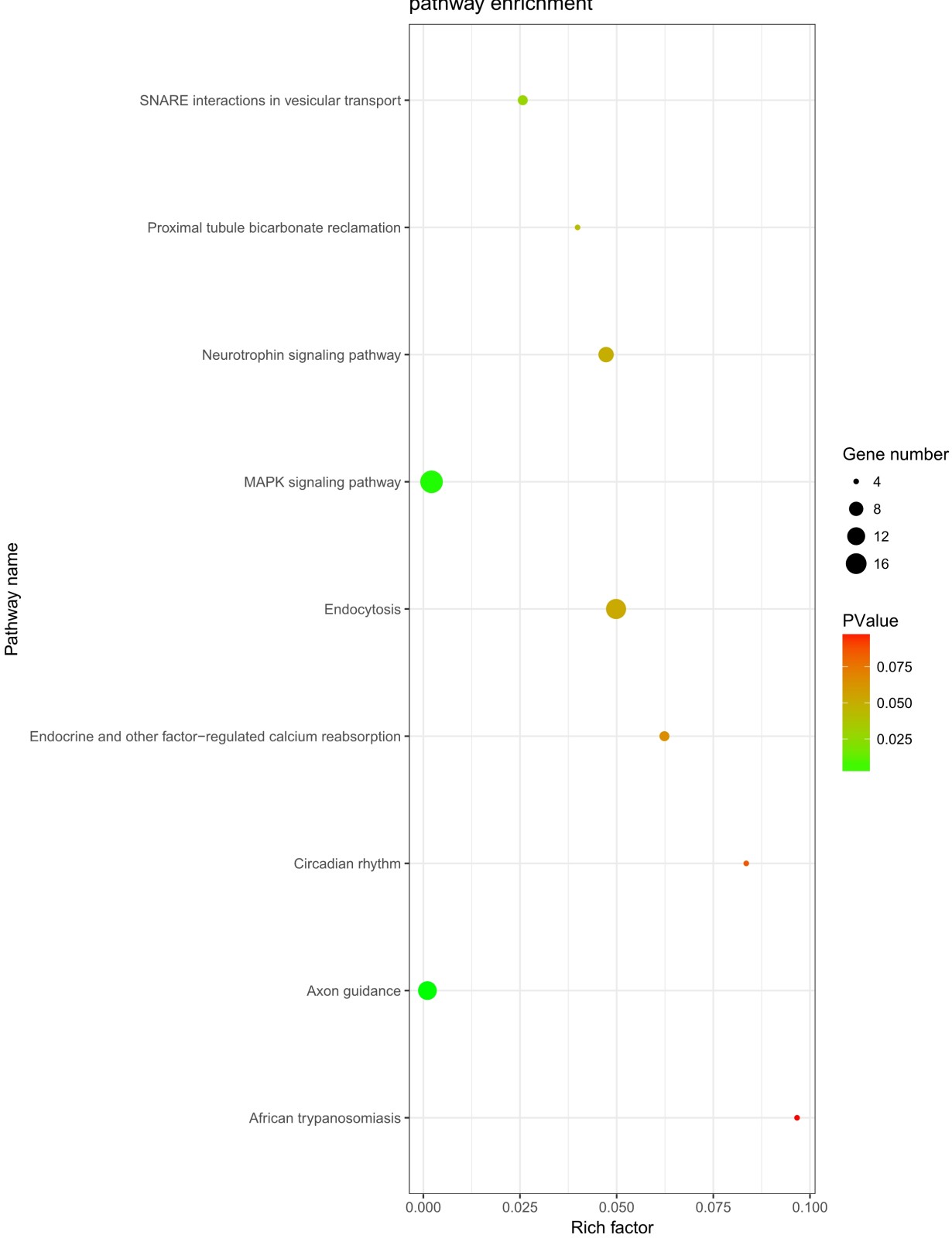

**Figure 5 KEGG analysis.** KEGG pathway analysis was conducted to determine the involvement of target genes in different biological pathways. The size of each circle indicates the number of circRNAs. The color of the circle indicates the *p*-value. The larger the circle and the lower of the *p*-value, the more enriched and meaningful the pathway.               

**Table 3 The top five (ranked by gene count) KEGG pathways.**

| Term | Count | *p*-value | Genes |
|---|---|---|---|
| MAPK signaling pathway | 19 | 0.002114 | PRKCA, FGF9, TAOK1, NLK, PPM1A, FASLG, PPM1B, STK4, SRF, RPS6KA5, RPS6KA6, MAP3K4, CRKL, PAK2, RASGRP3, RAP1A, MAPK8, RAPGEF2, DUSP6 |
| Endocytosis | 15 | 0.049837 | PARD6B, ERBB4, KIF5C, CBL, ASAP2, EEA1, SMAD2, PSD2, CLTC, ARPC3, IGF2R, KIAA1033, DNAJC6, RAB11A, WIPF1 |
| Axon guidance | 13 | 0.00105 | EPHA5, ABLIM1, EPHA4, NRP1, PAK2, SEMA6D, PLXNA2, ABLIM3, SEMA3D, SEMA3C, UNC5C, EPHB3, ITGB1 |
| Neurotrophin signaling pathway | 9 | 0.047261 | RPS6KA5, RPS6KA6, CRKL, RAP1A, SORT1, FASLG, MAPK8, IRS1, PTPN11 |
| SNARE interactions in vesicular transport | 5 | 0.025713 | STX5, VAMP4, BET1L, VAMP2, STX1B |
| Endocrine and other factor-regulated calcium reabsorption | 5 | 0.06235 | PRKCA, KL, RAB11A, CLTC, PLCB1 |
| Proximal tubule bicarbonate reclamation | 4 | 0.039887 | SLC38A3, GLS, GLUD1, PCK1 |
| Circadian rhythm | 4 | 0.083514 | CRY2, BHLHE40, RORA, FBXL3 |
| African trypanosomiasis | 4 | 0.096652 | PRKCA, FASLG, PLCB1, SELE |

hsa-miR-4311, and hsa-miR-875-3p. In turn, these miRNAs may be associated with 380 target genes, including *ZYX, PRKAR1A, BTAF1, LRP3*, and *ETS1*.

The potential regulatory role of circRNAs in radioresistance was further investigated by conducting KEGG and GO analyses and determining the differential expression of mRNAs in the two study groups. GO analysis was performed to annotate the BPs, CCs, and MFs of target genes. The GO analysis results indicated that the high expression of target genes might be involved in cell migration and angiogenesis. Many studies have shown that radiation can kill tumor cells and promote cell migration and invasion via different mechanisms. It was shown that sublethal X-ray can enhance cell migration (*Imaizumi et al., 2018*). Another study found that the higher was radioresistance, the higher was the expression of *VEGFR*, which might lead to fast angiogenesis (*Lee et al., 2017*). "Protein binding" was an important MF-related GO term associated with radiation. As for protein binding, it was reported that the specific binding of *VDR* and *p53* in irradiated HEK 293T was primarily related to cell fate decision (*Pemsel et al., 2018*).

Moreover, KEGG analysis indicated that the MAPK signaling pathway was the most enriched pathway, with 19 related genes. This classic pathway is highly conserved and is involved in many cellular functions, including cell proliferation, differentiation, and migration, which is consistent with the results of GO analysis. A study showed that baseline MAPK signaling activity conferred intrinsic radioresistance to KRAS-mutant colorectal carcinoma cells by rapid up-regulation of *hnRNP K* (*Eder et al., 2017*). Moreover, quiescent G0 cells were found to be more resistant to ionizing radiation than G1 cells because *P38 MAPK, phosphorylated P38 MAPK*, and *RAC2* were regulated in mutual feedback and negative feedback regulatory pathways, leading to the

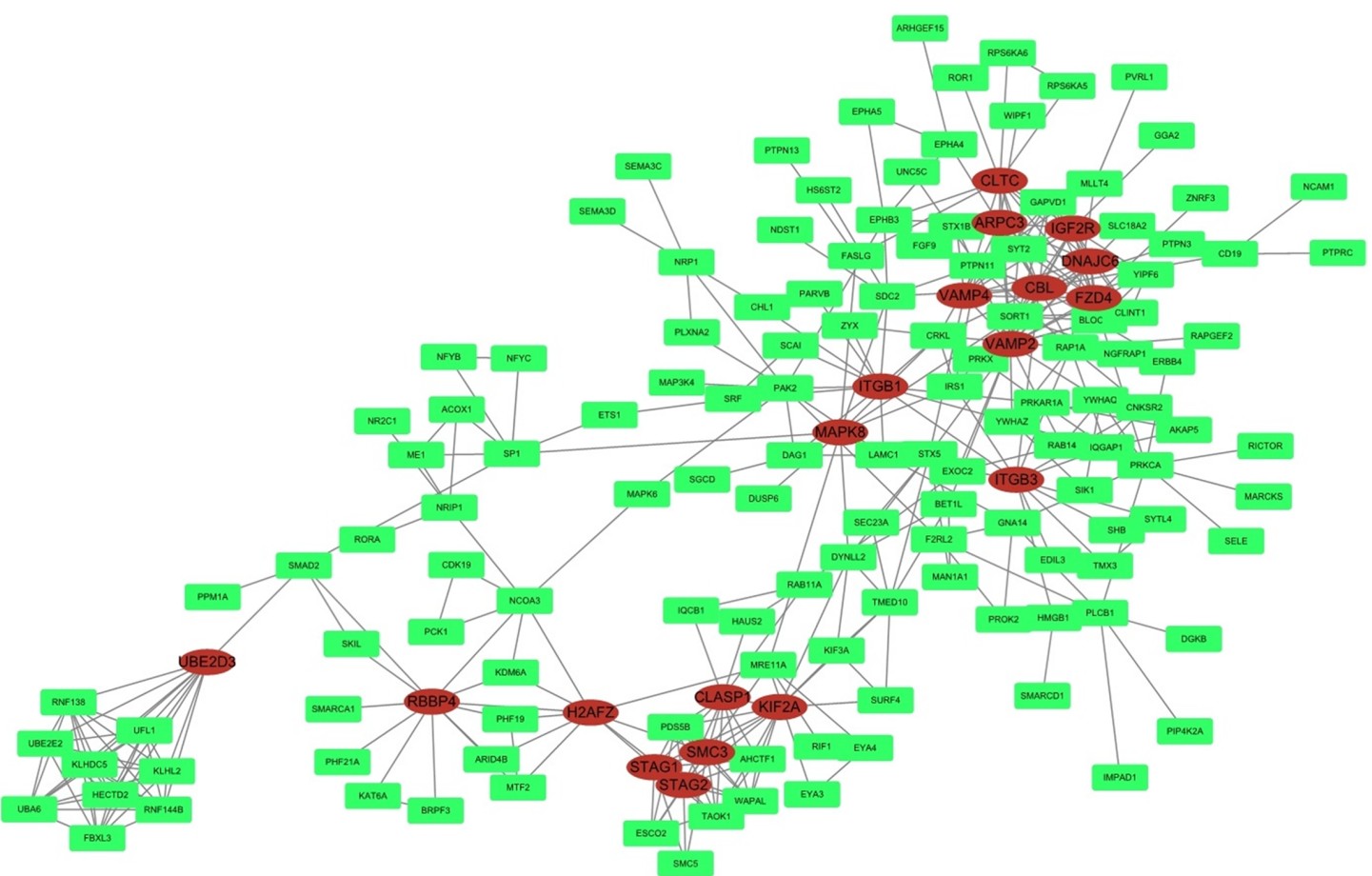

**Figure 6 PPI network.** STRING was used to predict protein interactions among the target genes. The interaction scores >0.9 (highest confidence) were selected for constructing PPI networks. The ovals represent proteins with >10 relationships with other proteins and may be the hub proteins in this network.

radioresistance of G0 cells (*Pei et al., 2017*). Interestingly, *RPS6KA5, RPS6KA6, CRKL, RAP1A, FASLG*, and *MAPK8* were found in both MAPK and neurotrophin signaling pathway. These genes may be involved in the crosstalk between different signaling pathways, but this hypothesis needs to be tested in further study.

Furthermore, in the PPI network analysis, 19 proteins were shown to associated with >10 other proteins and thus maybe considered as hub proteins in this network. Protein MAPK8 was associated with 10 other proteins and was also found in the MAPK and neurotrophin signaling pathway in KEGG analysis. Studies on this gene encoding MAPK8 suggest that this kinase plays a key role in T cell proliferation, apoptosis, and differentiation, and may be a crucial protein in cervical cancer radioresistance.

In summary, this study determined the profile of differentially expressed circRNAs and their target genes in HeLa cells subjected to radiation. Our results suggest that circRNAs may play a major role in the response to radiation. Considering that only a small percentage of the total circRNA population has been studied to date, the investigation of circRNAs is only beginning. Therefore, additional studies are needed to clarify the

importance of circRNAs in the response to radiation and may help elucidate the biological and molecular mechanisms underlying radioresistance in HeLa cells and allow the development of novel therapeutic approaches. The exploration of radioresistance-related circRNAs in radiation-treated HeLa cells may help provide responsive therapy in therapy-resistant cancer and improve prognosis.

## ACKNOWLEDGEMENTS

We thank Professor Jin Shunzi for guiding us about the bioinformatics.

### Funding

This work was supported by Technology Development Plan, Science and Technology Department of Jilin Province. The funders had no role in study design, data collection and analysis, decision to publish, or preparation of the manuscript.

### Grant Disclosures

The following grant information was disclosed by the authors:
Technology Development Plan, Science and Technology Department of Jilin Province.

### Competing Interests

The authors declare that they have no competing interests.

### Author Contributions

- Duo Yu performed the experiments.
- Yunfeng Li analyzed the data.
- Zhihui Ming analyzed the data, contributed reagents/materials/analysis tools.
- Hongyong Wang contributed reagents/materials/analysis tools.
- Zhuo Dong performed the experiments, prepared figures and/or tables, authored or reviewed drafts of the paper.
- Ling Qiu performed the experiments.
- Tiejun Wang conceived and designed the experiments, authored or reviewed drafts of the paper, approved the final draft.

### Data Availability

   Duo, Yu (2018): Raw Data. figshare. Fileset. https://doi.org/10.6084/m9.figshare.5802510.v1.

### Supplemental Information

Supplemental information for this article can be found online at http://dx.doi.org/10.7717/peerj.5011#supplemental-information.

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
