# Peer review of "Comprehensive circular RNA expression profile in radiation-treated HeLa cells and analysis of radioresistance-related circRNAs"

_PeerJ, doi:10.7717/peerj.5011_

## Round 0.1 · original submission · Major Revisions

As you can see there are serious problems with the current version. The main ones include the need of validation of the junction sequences of the circRNAs, need of grounding the assumptions about the genesis, levels and functions of the circRNA respect to linear RNAs.

Please respond to all the comments of the reviewers.

Reviewer 1 ·

Basic reporting

The central topic of the paper and objective of performed experiments is clear.

The authors provide sufficient background and context which makes the paper a clear read.

The article is self contained.

A significant number of tables, raw data, and experimental methods need to be additionally provided.

Experimental design

To my knowledge the paper is original with a clearly defined question.

Methodological the paper is not yet fully sound and it will require additions:

- Line 168: If samples were RNase R digested how was GAPDH used as an internal control.
- Line 181-183: the cut-off to define a circle should be stated (eg. number or samples with circle, number of counts per circle)
- Line 211: missing description of the small RNA sequencing
- Line 216: Different type of correlations should be mapped on the network figure.
- Line 259: "selected for" is probably a typo and should be corrected.
- Missing raw data for small RNA sequencing.
- Missing counts for circRNAs as well as small RNA sequencing results.
- Missing supplementary table for miranda results was well as circRNA vs miRNA correlations.

Validity of the findings

The findings are linked to the original research question and seem valid but need more supporting information to be robust as described in "2. Experimental design".

Reviewer 2 ·

Basic reporting

The text is poorly written and need English revision.

Experimental design

Need more experiments.

Validity of the findings

Findings are not experimentally supported.

Additional comments

In this manuscript by Yu et al., the authors have performed RNA sequencing to study the effects X-ray on circRNA using HeLa cells. They found 76 upregulated and 77 downregulated. Based on this data, they performed several bioinformatics analyses to conclude that circRNAs may play a role in radioresistance in Hela cells and allow novel therapeutic approaches.

Comments
Although authors have acquired RNA seq data with respect to circRNAs, the study remains very descriptive and premature. Validation of circRNAs by RT-qPCR needs to be thoroughly performed. Authors have validated only few downregulated circRNAs. Validation of more circRNAs will strengthen the manuscript. Also, validation the sequences of the PCR products is essential to confirm the existence and junction sequences of the circRNAs. The assumption that circRNA activity was related to known functions of host linear transcripts is not experimentally supported. First, the levels of circRNAs could be different from mRNAs. Second, the abundances of both circRNA and mRNAs may be differentially altered upon radiation. Third, the biogenesis of circRNAs is dependent on backsplicing (totally different from mRNAs). Thus, authors need to experimentally support this assumption since it is the ground stone of the rest of the manuscript and the bioinformatics analysis. In addition, authors use another assumption that circRNAs and miRNAs will be correlated. Again this is not experimentally supported. First, mRNA levels are not assessed in this study. Second, the levels of miRNAs do not reflect their activity, which is not determined in this study. Third, not all circRNAs will regulate miRNA activity. Finally, it does not seem that the study is representing clues about a possible role of circRNAs in radioresistance or novel therapeutic approaches.

Reviewer 3 ·

Basic reporting

English needs to be improved. For example:

line#27 Malignant tumors are less sensitive or do not respond to irradiation because
#31 We sought to investigate the role of circRNAs in radioresistance in Hela cells
#37 Results: RNA-Seq allowed
#52 high mortality of cervical cancer patients
#62 that compose

similarly, minor corrections are required at several places to improve the English quality of this manuscript

Experimental design

no comment

Validity of the findings

no comment

Additional comments

circRNAs research is becoming increasingly valuable in patient's diagnostics and potential therapeutic interventions. Here, the authors analyze the differentially regulated circRNA in HeLa cells, which confer radioresistance. With some careful corrections in language flow and English writing mistakes, the manuscript may be accepted.

Reviewer 4 ·

Basic reporting

The basic reporting of the findings is adequate given that English is most likely not the main language of the authors. The literature is not entirely complete and authors may consider the recent findings by O’Leary et al 2017 given the radiation responsive circRNA KIRKOS 73 is highly expressed in the human cervix

Experimental design

There are a number is issues however with the experimental design. As the main aim of this study appears to be finding a basis for radiotherapy resistance – why use X-ray irradiation and not gamma ray? 10 Gy is rather excessive and it is surprising that the cells survived this level. At what time point was the RNA extracted following X-ray exposure?
RNase R treatment conditions – 20 units for 6 mg RNA – 1 unit will treat 1ug or 20 units will treat 20 ug. The subsequent reliance on the sample as representative of circRNA is incorrect given the low levels of enzyme added to the total RNA pool.
Please explain the ‘blank control’ in more detail – is this non-irradiated or sham-irradiated?
Selecting 8 circRNAs at random for verifying the RNA-seq results is perhaps not the best way – the study would have been better placed to select the top 5 upregulated and top 5 downregulated circRNA from the RNA-seq data and confirm the findings with QPCR (and ideally automated sequencing for the presence of the back spliced junction). The ‘tendency’ of the findings to confirm the RNA-seq is not very convincing. Also, table 4 is not a table and perhaps could be put in instead of histograms where no comparison is provided at present.

Validity of the findings

Validity of the findings: The authors state that ‘GO and KEGG biological pathway analysis was performed to predict potential activity of circRNAs’ yet GOseq was used to analyse the potential of linear transcripts (line 125). The authors should be aware that circRNA is regulated independently of linear counterparts. The authors even state that circRNA show expression patterns that are different from their linear counterparts (Line 243-244).
Rather than a circRNA having many binding sites for one miRNA, more emphasis has been placed here on many miRNA potentially binding to a circRNA (the former scenario would have greater biological relevance).

While 16,893 circRNA were indicated as being identified (although the numbers didn’t quite add up lines 181- 183), only ~ 1% were differentially expressed between groups.

Additional comments

This study undertook RNA-sequencing of X-ray irradiated HeLa cells and standard bioinformatic analysis with the view to extrapolating a role for circRNA in radiotherapy resistance of cervical cancer. Using linear counterpart information an interpretation was made for potential function and pathway alterations by circRNAs. Circular RNA is a newly recognised class of non-coding RNA with varying opinions on its ability to act as sponges for microRNA. To date there is very limited knowledge of the exact function of circRNA. While the study provides a potential bioinformatic source of circRNA for future research, the manuscript tends to lean towards over-conclusion and lack of depth.

Minor points:
Keyword: function – replace with some other term more specific to the study.
Line 52: reword
There are a lot of words jointed together in error throughout the text eg.thatcompose
Line 68: sentence too short, please elaborate a bit more.
Line 76-77: are pseudogenes regarded as a competing RNA group?
HeLa cells not Hela cells.
Line 83: should be ‘and to explore’
Line 106: mass?
Line 116: ‘fold change of the same circRNA in both groups’ – please rephrase for more clarity.
Line 141: cor.test () ?
Line 152: change retro to reverse
Line 174-175: rephrase and add more detail.
Line 178: reword the second part of the sentence – too general.
Line 235: why italics? and please provide references.
Line 264: linear
Line 298: absolute values close to 1/-1 – please explain?

---

## Round 0.2 · accepted · Accept

Dear Authors,

Given that it was impossible to obtain reports from one of the original referees I have evaluated the new version and the rebuttal letter myself. I see that you have addressed the main criticisms and completed the missing aspects.

I can also see that even though it would be good to add experimental validation, adding more work can be considered out of scope for what is basically a methodological paper.

Based on this information together with the opinion of the first referee I can confirm that the article is now acceptable for publication.

Reviewer 4 ·

Basic reporting

The basic reporting has improved a lot upon revision.

Experimental design

This has been made clearer in the revised version.

Validity of the findings

The data is robust, statistically sound and is controlled correctly.

Additional comments

The manuscript is much improved.